# Phytase Supplementation of Growing-Finishing Pig Diets with Extruded Soya Seeds and Rapeseed Meal Improves Bone Mineralization and Carcass and Meat Quality

**DOI:** 10.3390/life13061275

**Published:** 2023-05-29

**Authors:** Anna Buzek, Anita Zaworska-Zakrzewska, Małgorzata Muzolf-Panek, Dagmara Łodyga, Dariusz Lisiak, Małgorzata Kasprowicz-Potocka

**Affiliations:** 1Department of Animal Nutrition, Faculty of Veterinary Medicine and Animal Science, Poznan University of Life Sciences, Wołyńska 33, 60-637 Poznań, Poland; anna.buzek@up.poznan.pl (A.B.); dagmara.lodyga@up.poznan.pl (D.Ł.); malgorzata.potocka@up.poznan.pl (M.K.-P.); 2Department of Food Quality and Safety Management, Faculty of Food Science and Nutrition, Poznan University of Life Sciences, Wojska Polskiego 31, 60-637 Poznań, Poland; malgorzata.muzolf-panek@up.poznan.pl; 3Department of Primary Meat Production, Institute of Agricultural and Food Biotechnology—State Research Institute, Głogowska 239, 60-111 Poznań, Poland; dariusz.lisiak@ibprs.pl

**Keywords:** pigs, phytase dosing, performance, fatty acid profile, meat quality, bone mineralization

## Abstract

The aim of this study was to determine how different doses of phytase in diets with extruded soybean seeds and rapeseed meal affected pigs’ growth performance, meat quality, bone mineralization, and fatty acid profiles. Sixty pigs were divided into three treatments by sex and body mass. Pigs were divided into starter (25 days), grower (36 days), and finisher (33 days) periods and fed with mash diets. No phytase was used in the control group diet, whereas in Phy1 and Phy2, 100 g and 400 g of phytase per ton of mixture were used, respectively. The feed conversion ratio and meat color were significantly correlated with phytase. Phytase supplementation had no effect on the growth of pigs, but total phosphorus was significantly increased in the bones and meat of the pigs. The enzyme additive reduced the C22:4 n-6 acid content in the meat, whereas other results were not significantly affected. The data suggest that the addition of phytase at a dosage of 100 g/ton to diets with extruded full-fat soya seeds and rapeseed meal can be valuable, as it reduces the FCR and increases the P content in the meat and bones.

## 1. Introduction

The main factor determining the profitability of pig production is feeding costs, which account for about 70% of the total variable costs [1]. Currently, soya is one of the most important sources of protein in animal nutrition worldwide, but about 80% of soybeans grown globally are GMOs [2]. In view of the growing aversion of the EU population towards genetically modified products and the need to secure the sources of materials for feed production in the event of an unexpected collapse in world trade, it is necessary to introduce and use other local high-protein components in animal nutrition [3]. Rapeseed and non-GM soybeans are important oilseeds cultivated in Europe. Both plant materials are rich in essential amino acids that complement each other (especially lysine and methionine); however, they are rarely used in unprocessed form in animal diets due to the high content of antinutritional factors, which can cause health problems in animals [4,5,6,7,8]. The nutritional value of raw feed components can be significantly improved by enrichment treatments such as extrusion, toasting, or extraction, as well as some enzyme additives. Heat treatments generally reduce thermolabile antinutrients, although the phytate content could also be altered during these processes. According to our results, extrusion reduced the content of phytate in soya seeds from about 0.5 to 0.3% (by 40%) [unpublished]. In turn, microbial exogenous phytase has the ability to hydrolyze phytate in feed, via a series of lower myo-inositol phosphate esters, into inositol [9]. Furthermore, enzymes such as phytase increase the energy value of feed, the digestibility of the protein, sugars, and fat from feed mixtures, and the availability of minerals from undigestible complexes [10,11]. In addition, phytase reduces the necessary complement of minerals from non-renewable sources to diets, reducing nutritional costs as well as phosphorus emissions into the environment [7,12,13]. Some studies have also shown that phytase at higher doses (above 2500 FTU/kg) results in a more favorable growth performance in pigs than standard phytase doses (usually levels well over 500 FTU/kg and up to 2500 FTU/kg) [14,15,16]. On the other hand, it has been found that dietary supplementation with high doses of phytase can have some beneficial effects on carcass quality parameters such as back-fat thickness in growing–finishing pigs [16]. Cauble et al. [17] and Gebert et al. [18] reported that dietary phytase could moderate muscle fatty acid profiles in chickens and pigs. Still, other authors need to prove those results. Additionally, some researchers found that exogenous phytase can influence serum non-esterified fatty acids and lipid metabolism in broilers and modulate the muscle fatty acid profile in shrimp [19,20]. Our research assumptions were that dietary phytase additives in fattener feed mixtures (1) are necessary in diets containing rapeseed meal (RSM) and extruded full-fat non-GMO soybeans (FFES), (2) are more effective at higher dosages (400 g/t), and (3) would enhance porker production and carcass results.

This study aimed to determine how a standard amount of phytase and a higher dosage added to a diet containing RSM and FFES influenced the performance of fatteners and their slaughter efficiency, as well as bone mineralization and meat quality.

## 2. Materials and Methods

### 2.1. Plant Material

RSM was purchased from the Bielmar Fat and Oil Processing Plant (Bielsko-Biała, Poland). Soybean seeds cv. Augusta (non-GMO) were obtained from Plant Breeding Stations (HR UP Poznań, Poland). Soybean seeds were extruded with a Farmet FE 250 extruder (Farmet as; Česká Skalice, Czech Republic). Preliminary crushing was applied to increase the efficiency of the machine. Then, the raw material was ground and heated. Under pressure and temperature, the material rich in fiber and protein was gradually softened and homogenized. The extrusion was conducted for 10 s at a temperature of 130 °C and a pressure of >20 MPa.

### 2.2. Enzyme

Phytase Quantum Blue 5G^®^ (AB Vista Feed, Marlborough, England) produced by *E. coli* was used in the experiments. The producer recommends that the conditioning of the feed mixture or granulation should not exceed a temperature of 90 °C and last longer than 30 s. The producer declares that the minimum phytase activity is 500 FTU/g with the recommended 100 g of the phytase additive per ton of mixture.

### 2.3. Ethical Statement

All experimental procedures complied with the guidelines of the Local Ethical Committee for Experiments on Animals in Poznań regarding animal experimentation and the care of animals under study (European Union (EU) Directive 2010/63/EU for animal experiments). Individual approval for this trial was not required because of the production standards used in this study. All samples were collected after slaughter. The pigs had unlimited access to feed and water.

### 2.4. Animals, Diets, and Experimental Design

The experiment was conducted on 60 castrated weaners (30 ♀ and 30 ♂) (Naima x Ebx) with an initial weight of about 31 kg. The animals were transported to the farm and divided into 3 groups in individual pens, each with 20 animals, with an even division of sex (n = 10). Each pig was marked with an earring with a number, which allowed individual measurements. The experiment lasted 94 days and was divided into three stages: starter (S), 25 days; grower (G), 36 days; and finisher (F), 33 days. RSM and FFES were the main sources of protein in the feed mixtures given to all groups of animals. No phytase was added to the mixture in the control group (Con). In the other two groups, namely Phy1 and Phy2, 100 g and 400 g of phytase per ton of mixture were, respectively, used in the diets. The diets for the test were formulated according to the recommendations of the GfE [21], as shown in Table 1. The health and welfare of the animals were monitored twice a day. After the completion of each fattening phase, the daily body weight gain (DBWG) of all animals was individually controlled. Due to the group housing of the animals, the ADFI and the feed conversion ratio (FCR) in each phase were estimated in the whole group. At the end of the experiment, 12 pigs from each group (6 males and 6 females) were stunned by electric shock and killed by exsanguination.

### 2.5. Analytical Procedures

#### 2.5.1. Feed Analyses

The feed mixtures were analyzed chemically twice (n = 2). A RetschZm 200 ultra-centrifugal mill (Retsch, Haan, Germany) with 1.0 mm sieves was used to grind the feed material. The material was analyzed for crude protein, crude fat, crude fiber, crude ash, and total Ca and P, using methods 976.05, 920.39, 978.10, 984.27, and 965.17, respectively, according to the AOAC [22].

#### 2.5.2. Carcass and Meat Analyses

The carcasses were measured (length and width (cm)) and weighed (kg), and the post-mortem yield (%) was calculated. On warm, hanging left carcass sides, the meatiness (%) and loin thickness (mm) were measured with an IM-03 ultrasound instrument. Additionally, the following parameters were measured: linear thickness of the back fat at three points (K I–K III: K I, less than 22 mm; K II, 22–26 mm; and K III, over 26 mm), the thickness of the back fat above the shoulder blade and at the last rib (mm), and the thickness of the buttock muscles and the loin (mm). Electrical conductivity in the muscle was measured on the left side of the carcass that was left hanging after 24 h of cooling. The water-holding capacity was analyzed using the Grau–Hamm method [23]. Drip loss and cooking loss were based on the differences in samples’ weights. Meat color was analyzed with a Minolta CR300 colorimeter. A group of panelists evaluated the cooked samples for tenderness. Muscle samples were analyzed twice for protein (method 990.03), dry matter (method 934.01), fat (method 920.39), and P (method 946.06) according to the methods of the AOAC [22].

#### 2.5.3. Bone Analysis

Third metacarpals from the right foot were collected from 12 pigs in each group. The metacarpals were boiled to remove tissues and cartilage caps, ground, and extracted to remove fat. Then, samples were burned in a muffle furnace (P330, Nabertherm GmbH, Lilienthal, Germany) at 600 °C for 5 h [24]. The P and Ca contents in the bone ash were determined according to procedures 984.27 and 965.17 of AOAC [22].

#### 2.5.4. Meat Fatty Acid Profile

The lipid fraction was extracted from 3 g of homogenized meat sample with 30 mL of Folch solution I (chloroform:methanol = 2:1, *v*/*v*). The homogenate was filtered with a Whatman No.1 paper filter. The samples were placed in 17 mL culture tubes, suspended in 2 mL of methanol, treated with 0.5 mL of 2 M aqueous sodium hydroxide, and sealed tightly. Then, the culture tubes were placed into 250 mL plastic bottles, sealed tightly, and placed inside a microwave oven (AVM 401/1WH, Whirlpool, Sweden) operating at 2450 MHz and 900 W maximum output. The samples were irradiated (370 W) for 20 s, followed by an additional 20 s after about 5 min. After 15 min, the contents of the culture tubes were neutralized with 1 M aqueous hydrochloric acid; 2 mL MeOH was added, and extraction with pentane (3–4 mL) was carried out in the culture tubes. The combined pentane extracts were evaporated to dryness in a nitrogen stream. Next, the extracts were methylated with a mixture of anhydrous methanol and sulfuric acid (1:5, *v*/*v*). Then, 0.5 mL of methanol was added to the extract containing lipids, followed by the addition of a mixture of 0.15 mL methanol/sulfuric acid (1:5, *v*/*v*). The samples were heated at 70 °C for 15 min. After the solution had been cooled, 0.5 mL of n-hexane was added, followed by the addition of a sufficient amount of water to form two layers. The upper hexane layer was removed and analyzed on a gas chromatograph (Agilent 5890 II) equipped with a flame ionization detector, fitted with a Supelcowax 10 column (30 m × 0.25 mm I.D., 0.25 mm film thickness). The injector and detector temperatures were 220 °C and 240 °C, respectively. The column temperature was programmed to increase from 60 °C to 240 °C at a rate of 110 °C/min. Peaks were identified by comparing the sample peak retention times with those of known methylated fatty acid compounds. Additionally, the determined contents of the fatty acid profiles allowed us to compute the following: thrombogenicity index (TI), saturated fat index (S/P), consumer index (CI), desaturation ratio from oleic to linoleic acid (ODR), desaturation ratio from linoleic to linolenic acid (LDR), calculated oxidizable value (COX), index of desirable fatty acids (DFA), and sum of hypercholesterolemic fatty acids (OFA). The formulas for the indicators of the nutritional quality of fat are presented in Table 2.

#### 2.5.5. Statistical Analysis

The SAS ver. 5.0. software (IO, Cary, NC, USA) was used for statistical analysis. The hypotheses were tested at α = 0.05. Differences were considered significant at *p* < 0.05. The obtained results were analyzed statistically by calculating the arithmetic mean and ± SD for each characteristic. The significance of differences between the groups in the experiment on pigs was calculated using one-way ANOVA with Duncan’s post hoc test. The correlation between FI and the phytase additive was tested with Spearman’s rank correlation coefficient (P. R). Principle component analysis (PCA) was used to visualize the information and to detect some patterns in the dataset. The clustering method (CA) using k-means clustering analysis was used to define distinct groups within the dataset with the lowest variability within the group and the highest variability between groups. To this end, the Euclidean distance was used as a metric between centroids. The number of clusters was defined automatically using V-fold cross-validation. The method constructs clusters using two goals, namely minimizing the variability within clusters and maximizing the variability between clusters. General discriminant analysis (GDA) and the supervised pattern recognition method proposed by Berrueta et al. [27] were used to calculate classification rules for sample discrimination. Before analysis, the data were pre-processed via a logarithmic transformation, and the sum-in-row approach was applied in order to obtain more homogenous variance in the dataset.

## 3. Results

The experiment did not reveal any negative effect of the administered mixtures on the animals’ health or welfare. No lameness, hernias, diarrhea, bitten tails, ear wounds, wounds on the body, ectoparasites, or swollen joins were noticed.

### 3.1. Performance Parameters

The fattening results are shown in Table 3. The enzyme additive did not increase the weight gain in the entire experiment (*p* > 0.05). However, the DBWG tended to increase along with the enzyme dose (*p* = 0.100). The total FCR in both experimental groups was significantly lower in comparison with that of Con (*p* < 0.05). The phytase dosage was significantly correlated with the FI values, as evidenced by Spearman’s correlation coefficient (R = −0.51; *p* < 0.01).

### 3.2. Carcass and Meat Quality

There were no significant differences in the carcass parameters among the tested groups (Table 4) (*p* > 0.05). The enzyme additive had no effect on the evaluation of the post-mortem parameters (*p* > 0.05).

Table 5 shows the chemical composition of the muscle meat. No significant effect was detected for meat quality in pigs fed diets with phytase compared to pigs fed diets without phytase. However, phytase significantly increased (*p* < 0.01) the muscle contents of P compared to pigs fed diets without phytase. One-way ANOVA only showed a tendency (*p* = 0.054) of phytase to impact the pork’s yellow color (b*).

### 3.3. Fatty Acid Profile

The fatty acid profile showed that the enzyme additive reduced only the content of the C22: 4n6 acid (*p* = 0.008), compared to the Con group. However, the groups did not differ significantly in the share of other acids (*p* > 0.05). The results indicate that microbial phytase did not cause changes in the fatty acid profile of the lipid fractions of the longissimus dorsi muscle. The total MUFA amounted to about 47%, whereas the total SFA and PUFA amounted to about 39% and 14%, respectively (Table 6).

One-way ANOVA did not show any significant effect of the phytase dose on the measured indices (Table 7).

Multivariate analyses (PCA, CA, GDA) of the fatty acid contribution were also applied. PCA was applied to investigate the structure of relationships between the variables (fatty acid contribution). Six principal components (PCs) showed an eigenvalue greater than 1, and together they explained 80% of the total variance, where the first two PCs explained 57% of the total variance. Figure 1 shows the projection of the variables on the factor plane defined by the two PCs (PC1 and PC2). PCA was applied to fatty acids in this study. It showed that C18:1 was highly correlated with the total MUFA, whereas C16:0 was correlated with the total SFA. PUFAs were highly correlated with C18:2 n-6. This is in accordance with the results shown in Table 5. All those variables distributed the samples along the PC1 axis, with loaded values ranging from −0.90 for C18:2 n-6 to −0.95 for SFA and PUFA. C20:4 n-6 and Σn-3 distributed the samples along the PC2 axis, with loaded values of −0.70 and 0.93, respectively.

However, PCA is not designed as a typical clustering method. To differentiate samples, k-means clustering analysis was applied to form clusters that were as distinct as possible. The method enables the distinction of four clusters. All samples subjected to the phytase treatment of 400 g/t (Phy2) were included in cluster 2. All control samples were included in cluster 1. Clusters 3 and 4 included samples treated with phytase at 100 g/t (Phy1). The results are shown in Figure 2.

Additionally, GDA was applied to discriminate groups in the sample dataset. Using both discriminant functions (Roots 1 and 2), 100% of the classification property of the phytase dose could be obtained with fatty acids as predictors (Figure 3). The canonical correlation coefficient for Root 1 was equal to 0.90, with *p* = 0.021.

### 3.4. Pig Metacarpal Bone Characteristics

The contents of ash, Ca, and P in the bones are shown in Table 8. The factor under study did not significantly influence the bone ash or calcium content; however, there was a significantly higher P content in the bones of the pigs from both groups fed with the enzyme additive (*p* < 0.0001).

## 4. Discussion

The effect of phytase depends strongly on the content of phytate in the components. Raw soybeans contain about 0.4% of phytic phosphorus, and some research has shown that its content is reduced to about 0.05–0.06% by extrusion [6]. Cereal seeds as well as RSM are generally rich in phytate. As they make up more than 80% of feed mixtures for pigs, the addition of phytase may be justified. However, the use of a recommended phytase dose and also its higher dosing in this study did not significantly affect the fattening results. Only the FCR in the whole experiment was improved, and the DBWG tended to increase when pigs were offered phytase. Wiśniewska et al. [11] and Wu et al. [28] found a positive correlation between percentage improvements in feed efficiency in response to phytase and dietary phytate content, including for diets based on legume seeds, SBM, or RSM [29,30]. Grela et al. [31] observed that a dietary phytase treatment at doses of 500–1000 FTU per kg significantly influenced daily gains and the FCR, but the FI was not affected. Other studies showed that phytase enhanced the utilization of some nutrients, but it was not always connected with increased performance [29,32]. It is commonly known that the effect of microbial phytase also depends on the source of phytate, the animal’s species and age, concentrations of minerals in the diet, phytase sources, and the phytase dosage [10,32]. Zeng et al. [33] found that, in comparison with the 500 FTU/kg inclusion group, super-high phytase doses (20,000 FTU/kg) increased daily gains as well as the digestibility of Ca and P. Our study found that phytic phosphorus could be a limiting factor in full-fat soybean seeds and rapeseed meal, as no growth-promoting effect was found due to the phytase addition. In this study, the lower FCR in the whole experiment was probably the result of higher energy availability due to the decomposition of phytate complexes with some nutrients. This was also proved by the significantly higher deposition of phosphorus in the bones of pigs offered the enzyme [33]. The correct phosphorus and calcium levels in the diet are necessary for bone mineralization. In our study, phytase had no effect on the contents of ash and calcium, but it significantly increased the phosphorus content in the bones and meat. This ability increases with the dose of the enzyme, as reported by other authors [10,34]. Grela et al. [31] found that the administration of 500 FTU of phytase/kg of diet increased bone strength. Kasprowicz-Potocka et al. [7] found that the phytase additive improved the availability of P and Ca in pigs’ diet by 14.34% and 4.08%, respectively. Some authors suggested that a higher dosing of phytase increases the digestible nutrient intake by stimulating feed intake, because phytate might act as an appetite suppressant [13]. Simultaneously, there was no effect of the enzyme on the carcass parameters and meat quality, except for the yellow color (b) of the meat. Hollowey et al. [35] found that a higher phytase dosing affected the growth rate, feed intake, and carcass yield (*p* > 0.10). Gebert et al. [18] offered pigs 1200 FTU of phytase/kg of feed and found that the slaughter yield, the percentage of lean cuts, and the total fat tissue as well as the back-fat thickness were not affected by phytase. Phytase supplementation only affected the meat color brightness and texture, which was also found in our results. The meat color is influenced by various factors such as the pigment content (heme proteins, especially myoglobin), the chemical condition of myoglobin (oxidized form of heme iron), and the rate of glycolysis in the meat. The addition of phytase increases the availability of trace elements such as iron and copper, which probably catalyze the autoxidation of myoglobin by stimulating the formation of hydrogen peroxide, the initial product of lipid oxidation, which affects the meat color. Moreover, the low-ultimate-effect myoglobin is readily oxidized to metmyoglobin, which does not contribute greatly to the depth of color; furthermore, the structure of the muscle is “open” and scatters light [17].

The lipid fraction in pig diets derives mostly from cereals (wheat, triticale, or maize) and processed vegetable oils or oilseed plants, which are rich in n-6 PUFA and have high n-6/n-3 ratios, which are linked to gut and metabolic inflammation. In our study, the enzyme additive only reduced the content of docosatetraenoic acid in the meat, but the mechanism was not clear. It is known that the C18:3n6/C22:4n6 ratio is a good lipid marker of chronic kidney disease (CKD) progression [36]. A reduction in the C22:4n6 content involves an increase in this ratio, which is beneficial for animal and human health. The C18:1 content was highly correlated with the total MUFA, whereas C16:0 was correlated with the total SFA. C16:0 has been found to raise low-density lipoprotein cholesterol; therefore, a reduced SFA level in the meat can improve dietetic and health-promoting meat properties. Sońta et al. [37] claimed that comparative analyses concerning the nutritionally important fatty acids in the meat from pigs offered feed mixtures containing protein from various plant sources, including legumes, are needed. The major PUFAs were C18:2 n6 and C18:3 n3. In the current research, PUFAs were highly correlated with C18:2 n-6, and they have previously been found to prevent coronary heart disease. The level of C18:2 n6 is associated with a lower amount of endogenous lipids in the meat. This phenomenon has a negative impact on the fat tissue during processing, as it lowers the melting temperature. Cauble et al. [17] found that dietary phytase can be a potential modulator of muscle fatty acid profiles, which was only slightly observed in the current study. The PCA showed that, in comparison with the Con group, the samples with a higher dosage of phytase were characterized by a relatively high SFA contribution to the total fatty acid content. Gebert et al. [18] also found increased SFA and reduced MUFA contents in microbial phytase-supplemented diets. These authors claimed that this may have been caused by the fact that microbial phytase indirectly made some slight changes in the fatty acid profile of the lipid fractions of the *longissimus dorsi* muscle. It is likely that the effect of dietary phytase on SFA was mainly caused by the higher feed intake and thus resulted in a higher self-synthesis of SFA or that the higher percentage of SFA in the complex lipids in the phytase-supplemented groups was a consequence of lipid oxidation. The fatty acids were reduced to secondary products, or they were saturated to less unsaturated fatty acids, lipid hydroperoxides, or aldehydes [38]. Biswas et al. [20] reported no significant differences in fatty acid profiles when rainbow trout were fed with or without phytase added to a soybean-meal-based diet. The values of the nutritional quality indices of fat determined in the meat of fatteners were the same as those of the Con group and are beneficial from a dietetic point of view.

## 5. Conclusions

In summary, adding phytase to the primary dose may benefit diets containing processed protein raw materials such as rapeseed meal and full-fat soybean seeds. However, it did not enhance the performance of fattening pigs, nor did it result in higher carcass quality, meat quality, or fatty acid indices. Only the feed conversion ratio, meat color (yellow), and bone and meat phosphorus contents were significantly correlated with the phytase additive. Moreover, the observed reduction in the C22:4n6 content led to an increase in the C18:3n6/C22:4n6 ratio, which could be beneficial for animal health. Our findings also suggest that the Ca deposition in the bones was not affected by the phytase content in this type of diet. Moreover, in the current study, the phytase additive was given “on top”, similar to most practices on pig farms. This research could be improved by lowering the content of phosphate in the diet, which could slightly reduce diet costs.

## Figures and Tables

**Figure 1 life-13-01275-f001:**
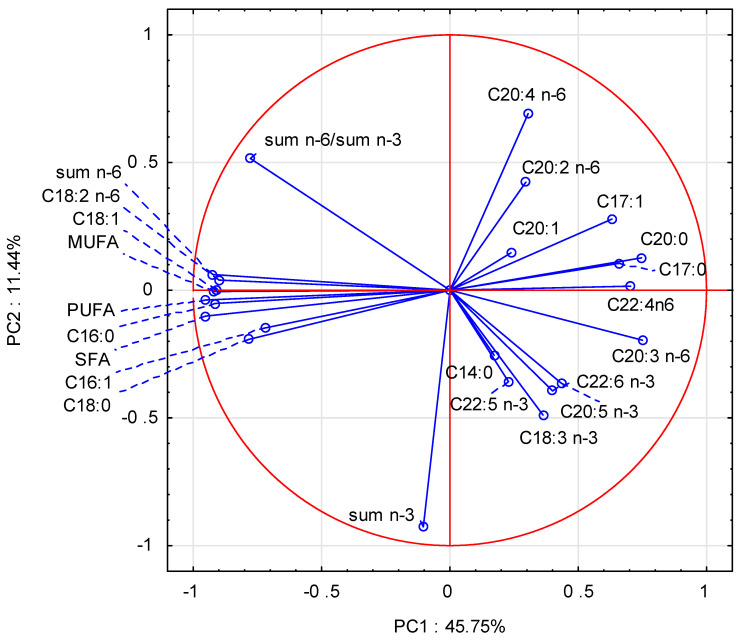
A projection of the fatty acid contribution on the factor plane, PC1 vs. PC2.

**Figure 2 life-13-01275-f002:**
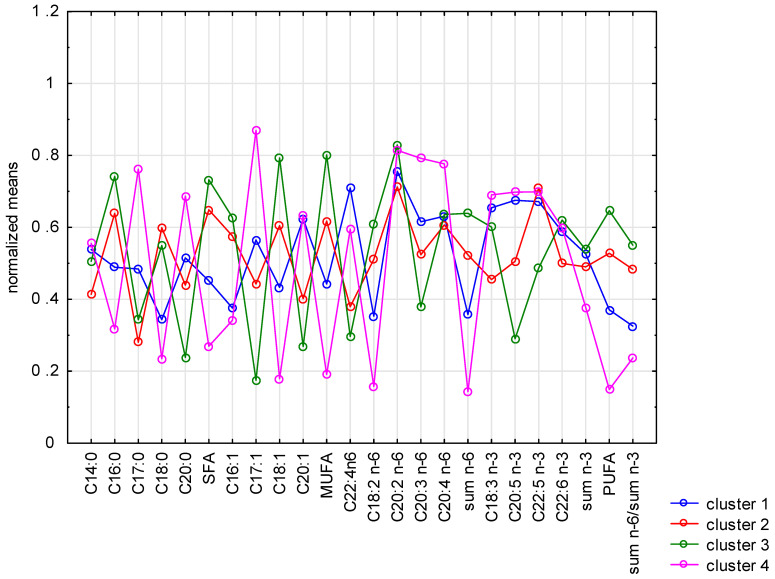
The normalized mean values of the fatty acid contribution for each cluster calculated using k-means clustering analysis.

**Figure 3 life-13-01275-f003:**
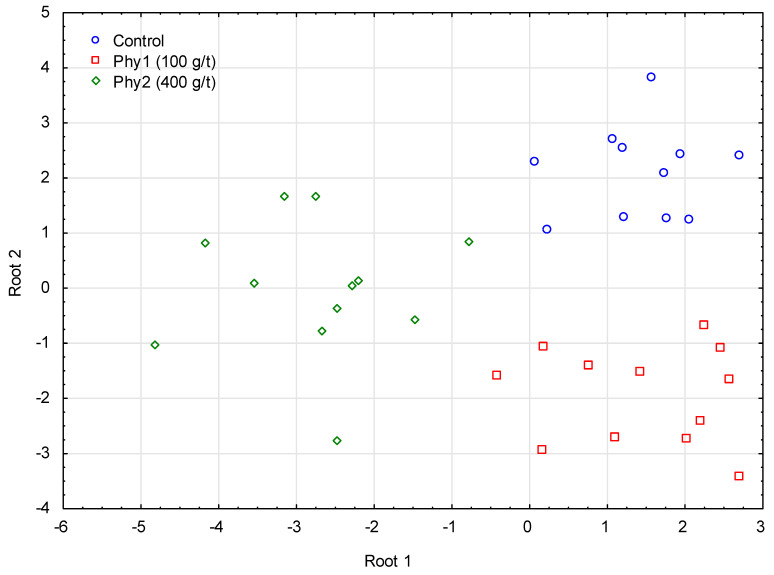
GDA classification of the samples by the phytase dose, with the fatty acid contributions as predictors. Con—control group; Phy1—phytase added at a dose of 100 g/t; Phy2—phytase added at a dose of 400 g/t.

**Table 1 life-13-01275-t001:** The composition and nutritional value of the diets in the starter, grower, and finisher stages.

Components (%)/Period	Starter	Grower	Finisher
Wheat	25.00	-	-
Maize	20.27	44.65	40.57
RSM	11.50	15.00	13.00
FFES	10.00	7.50	5.00
Rye	10.00	20.00	25.00
Barley	10.00	-	-
Wheat bran	9.50	10.00	14.00
Limestone	1.45	1.40	1.30
1-Calcium phosphate	0.30	0.25	-
Sodium chloride	0.40	0.40	0.40
Premix grower 0.2% *	0.20	0.20	0.15
Magnesium oxide	0.20	0.20	0.20
HCL-lysine 78.5%	0.51	0.36	0.36
DL-methionine 99%	0.22	-	-
L-threonine 98.5%	0.03	0.02	-
L-tryptophan 98%	0.03	-	-
Liquid acidifier **	0.35	-	-
Choline chloride	0.04	0.02	0.02
Calculated nutritional value (g/kg)			
Crude protein	170.32	171.60	153.20
Crude fat	41.70	41.20	34.20
Crude fiber	48.84	55.50	48.80
Ca	8.29	6.10	6.70
P	5.85	5.30	4.90
Ca/P	1.2–1.4: 1		

The table shows the composition and nutritional value of the mixtures of the control group (Con). In the experimental groups Phy 1 and Phy 2, 100 and 400 g/kg, respectively, of enzyme phytase Quantum Blue 5G were added to the feed mixture, replacing the proportion of maize. RSM—rapeseed meal; FFES—full-fat extruded soybeans. * The mineral and vitamin premix contained the following amounts of components per 1 kg: vitamin A—3,250,000 IU/kg; vitamin D_3_—1,000,000 IU/kg; vitamin E—50,000 mg/kg; vitamin K_3_—2000 mg/kg; vitamin B_1_—1000 mg/kg; vitamin B_2_—2000 mg/kg; vitamin B_6_—1500 mg/kg; vitamin B_12_—15 mg/kg; pantothenic acid—1500 mg, 5000 mg/kg; nicotinic acid—1000 mg/kg; biotin—50 mg/kg; folic acid—750 mg/kg; Fe—50,000 mg/kg; Mn—37,000 mg/kg; Zn—50,000 mg/kg; Cu—10,000 mg/kg; I—750 mg/kg; Se—200 mg/kg. ** The liquid acidifier contained formic acid, propionic acid, ammonium formate, ammonium propionate, demineralized water, glycerine, and glycol.

**Table 2 life-13-01275-t002:** The indicators of the nutritional quality of fatty acids.

Indices	Formula	References
TI	$\frac{C14:0+C16:0+C18:0}{0.5*MUFA+0.5*n6PUFA+3*n3PUFA+(\frac{n3}{n6})}$	Szpunar-Krok et al. [25]
S/P	$\frac{C14:0+C16:0+C18:0}{\left(MUFA+PUFA\right)}$	
OL/(LA + ALA)	$\frac{C18:1}{C18:2+C18:3}$	
CI	C18:3+C20:5+C22:6	
ODR	$\frac{C18:2+C18:3}{C18:1+C18:2+C18:3}*100$	Mondal et al. [26]
LDR	$\frac{C18:3}{C18:2+C18:3}*100$	
COX	$\frac{1*(C16:1+C17:1+C18:1+C20:1)+10.3*C18:2+21.6*(C18:3+C20:3)}{100}$	Szpunar-Krok et al., 2022 [25]
DFA	C18:0+∑UFA	
OFA	C16:0+C14:0	

Con—control group; Phy1—control group + phytase 100 g/t; Phy2—control group + phytase 400 g/t; TI—thrombogenicity index; S/P—saturated fat index; OL—oleic acid (C18:1); LA—linoleic acid (C18:2); ALA—alpha linoleic acid (C18:3); CI—consumer index; ODR—desaturation ratio from oleic to linoleic acid; LDR—desaturation ratio from linoleic to linolenic acid; COX—calculated oxidizable value; DFA—index of desirable fatty acids; OFA—sum of hypercholesterolemic fatty acids.

**Table 3 life-13-01275-t003:** The pigs’ performance parameters.

Phase/Variable	Con	Phy1	Phy2	SEM	*p*
Starter					
IBW (kg)	30.84 ± 1.04	31.33 ± 0.98	31.35 ± 1.02	0.392	>0.05
FBW (kg)	51.37 ± 6.87	53.20 ± 8.07	53.80 ± 6.08	0.910	>0.05
DBWG (kg)	0.821 ± 0.13	0.875 ± 0.16	0.898 ± 0.13	0.018	>0.05
FCR (kg/kg)	2.38 ± 0.42	2.32 ± 0.47	2.25 ± 0.35	0.050	>0.05
Grower					
FBW (kg)	90.58 ± 12.08	95.40 ± 12.37	97.28 ± 11.06	0.562	>0.05
DBWG (kg)	1.09 ± 0.21	1.17 ± 0.18	1.21 ± 0.14	0.024	>0.05
FCR (kg/kg)	2.60 ± 0.61	2.43 ± 0.40	2.38 ± 0.29	0.588	>0.05
Finisher					
FBW (kg)	124.05 ± 15.31	131.35 ± 13.98	131.55 ± 12.86	1.860	>0.05
DBWG (kg)	1.01 ± 1.20	1.09 ± 0.14	1.04 ± 0.16	0.022	>0.05
FCR (kg/kg)	4.97 ± 0.61	4.04 ± 0.40	4.31 ± 0.29	0.200	>0.05
Total					
BWG (kg)	93.21 ± 12.96	100.03 ± 11.17	100.20 ± 9.71	1.510	>0.05
DBWG (kg)	1.00 ± 0.14	1.08 ± 0.12	1.08 ± 0.10	0.016	>0.05
FCR (kg/kg)	3.25 ^a^ ± 0.51	2.96 ^b^ ± 0.46	2.98 ^b^ ± 0.31	0.054	0.049

Con—control group; Phy1—control group + phytase 100 g/t; Phy2—control group + phytase 400 g/t; IBW—initial body weight; FBW—final body weight; DBWG—daily body weight gain; BWG—body weight gain; FCR—feed conversion ratio. ^a^,^b^ Means with different superscripts in a row are different *(p* ≤ 0.05).

**Table 4 life-13-01275-t004:** The carcass characteristics of growing pigs.

Phase/Variable	Con	Phy1	Phy2	SEM	*p*
Carcass weight (kg)	97.91 ± 12.94	102.24 ± 11.71	102.83 ± 11.01	1.550	>0.05
Meatiness (%)	57.30 ± 3.06	57.64 ± 2.55	56.35 ± 2.31	0.350	>0.05
Cold dressing yield (%)	78.85 ± 1.35	77.80 ± 1.46	78.14 ± 1.13	0.240	>0.05
Mean thickness of pork loin (mm)	66.90 ± 7.30	66.10 ± 6.30	66.30 ± 6.50	0.900	>0.05
Mean back-fat thickness (mm)	17.32 ± 4.56	16.62 ± 3.61	18.72 ± 3.18	0.500	>0.05
pH 45 min	6.473 ± 0.22	6.57 ± 0.23	6.45 ± 0.30	0.040	>0.05
Back-fat thickness K III (mm)	20.50 ± 6.16	20.50 ± 6.46	22.84 ± 5.64	1.000	>0.05
Back-fat thickness K II (mm)	17.75 ± 5.66	16.67 ± 5.76	18.42 ± 4.17	0.858	>0.05
Back-fat thickness K I (mm)	24.67 ± 5.55	23.00 ± 6.72	25.25 ± 3.33	0.886	>0.05
GMT (mm)	72.67 ± 7.11	71.42 ± 7.14	74.17 ± 7.16	1.170	>0.05
Back (cm)	21.00 ± 5.17	20.00 ± 3.46	22.17 ± 3.88	0.700	>0.05
Shoulder (cm)	43.08 ± 7.82	43.67 ± 6.54	47.00 ± 6.19	1.150	>0.05
Carcass length (cm)	89.67 ± 3.28	89.08 ± 3.03	88.92 ± 3.26	0.520	>0.05
Carcass width (cm)	39.34 ± 1.15	38.08 ± 2.35	38.67 ± 1.15	0.280	>0.05
EC (mS/cm)	4.54 ± 0.76	4.79 ± 1.20	4.32 ± 1.13	0.170	>0.05

Con—control group; Phy1—control group + phytase 100 g/t; Phy2—control group + phytase 400 g/t; GMT—gluteal muscle thickness; EC—electrical conductivity of muscles. Back-fat thickness was measured at 3 points (K I–K III: K I, less than 22 mm; K II, 22–26 mm; and K III, over 26 mm).

**Table 5 life-13-01275-t005:** Meat quality parameters from *m. longissimus lumborum*.

Variable	Con	Phy1	Phy2	SEM	*p*
Fat (%)	1.81 ± 0.57	2.31 ± 0.59	2.19 ± 0.57	0.101	>0.05
Protein (%)	24.21 ± 0.84	23.54 ± 1.30	24.21 ± 0.66	0.166	>0.05
Water (%)	72.76 ± 0.77	72.82 ± 0.80	72.42 ± 0.84	0.134	>0.05
Dry matter (%)	27.24 ± 0.77	27.18 ± 0.80	27.59 ± 0.84	0.134	>0.05
P (%)	1.05 ^b^ ± 0.04	1.19 ^a^ ± 0.04	1.19 ^a^ ± 0.03	0.01	<0.001
pH 24 h	5.53 ± 0.08	5.53 ± 0.12	5.48 ± 0.10	0.017	>0.05
L*—lightness	46.83 ± 1.81	46.42 ± 2.83	47.86 ± 3.37	0.457	>0.05
a*—redness	5.12 ± 0.76	5.48 ± 0.96	5.57 ± 1.14	0.160	>0.05
b*—yellowness	1.96 ± 1.13	1.79 ± 0.90	2.84 ± 1.15	0.191	>0.05
Cooking loss (%)	26.38 ± 5.19	26.12 ± 4.22	28.16 ± 4.34	0.760	>0.05
Tenderness (%)	27.09 ± 5.07	26.04 ± 8.40	26.31 ± 4.26	1.000	>0.05
Drip loss (%)	5.21 ± 1.74	5.10 ± 1.39	6.15 ± 1.21	0.250	>0.05
Water-holding capacity (%)	28.07 ± 4.16	28.26 ± 1.81	28.36 ± 2.93	0.510	>0.05

Con—control group; Phy1—control group + phytase 100 g/t; Phy2—control group + phytase 400 g/t. ^a^,^b^ Means with different superscripts in a row are different *(p* ≤ 0.05).

**Table 6 life-13-01275-t006:** The fatty acid profile (% of total FA) of the fat from *m. longissimus lumborum*.

Variable	Con	Phy1	Phy2	SEM	*p*
C14:0	1.32 ± 0.14	1.32 ± 0.14	1.26 ± 0.09	0.021	>0.05
C16:0	23.78 ± 0.96	23.68 ± 1.04	23.67 ± 0.84	0.154	>0.05
C17:0	0.22 ± 0.02	0.23 ± 0.01	0.21 ± 0.01	0.003	>0.05
C18:0	13.36 ± 1.02	13.27 ± 1.04	13.72 ± 0.78	0.157	>0.05
C20:0	0.21 ± 0.01	0.21 ± 0.01	0.22 ± 0.01	0.002	>0.05
C16:1	2.32 ± 0.10	2.36 ± 0.09	2.37 ± 0.09	0.016	>0.05
C17:1	0.12 ± 0.02	0.12 ± 0.02	0.12 ± 0.02	0.003	>0.05
C18:1	43.59 ± 1.72	43.77 ± 1.70	43.501 ± 1.22	0.254	>0.05
C20:1	1.12 ± 0.04	1.09 ± 0.05	1.08 ± 0.06	0.008	>0.05
C22:4 n-6	0.12 ^b^ ± 0.01	0.11 ^a^ ± 0.01	0.11 ^a^ ± 0.01	0.001	0.008
C18:2 n-6	10.89 ± 0.66	10.90 ± 0.63	10.86 ± 0.51	0.098	>0.05
C20:2 n-6	0.33 ± 0.06	0.34 ± 0.07	0.33 ± 0.03	0.009	>0.05
C20:3 n-6	0.21 ± 0.08	0.21 ± 0.06	0.20 ± 0.06	0.011	>0.05
C20:4 n-6	0.14 ± 0.04	0.15 ± 0.04	0.13 ± 0.02	0.005	>0.05
C18:3 n-3	0.92 ± 0.05	0.92 ± 0.06	0.89 ± 0.05	0.009	>0.05
C20:5 n-3	0.42 ± 0.06	0.38 ± 0.07	0.39 ± 0.07	0.011	>0.05
C22:5 n-3	0.44 ± 0.06	0.42 ± 0.05	0.45 ± 0.04	0.008	>0.05
C22:6 n-3	0.49 ± 0.05	0.51 ± 0.06	0.49 ± 0.06	0.009	>0.05
SFA	38.91 ± 1.69	38.72 ± 1.41	39.08 ± 1.31	0.240	>0.05
MUFA	47.14 ± 1.70	47.35 ± 1.62	47.06 ± 1.25	0.249	>0.05
PUFA	13.95 ± 0.66	13.94 ± 0.65	13.85 ± 0.46	0.097	>0.05
Σ n-3	2.28 ± 0.10	2.24 ± 0.12	2.22 ± 0.12	0.019	>0.05
Σ n-6	11.77 ± 0.67	11.70 ± 0.58	11.64 ± 0.51	0.096	>0.05
Σ n-6/Σ n-3	5.14 ± 0.42	5.20 ± 0.29	5.26 ± 0.44	0.060	>0.05

Con—control group; Phy1—control group + phytase 100 g/t; Phy2—control group + phytase 400 g/t; SFA—saturated fatty acids; MUFA—monounsaturated fatty acids; PUFA—polyunsaturated fatty acids. ^a^,^b^ Means with different superscripts in a row are different (*p* ≤ 0.05).

**Table 7 life-13-01275-t007:** Values of the nutritional quality indices of the fat.

Indices	Con	Phy1	Phy2	SEM	*p*
TI	1.06 ± 0.07	1.05 ± 0.06	1.07 ± 0.06	0.061	>0.05
S/P	0.63 ± 0.05	0.63 ± 0.04	0.64 ± 0.04	0.039	>0.05
OL/(LA + ALA)	3.70 ± 0.27	3.72 ± 0.32	3.71 ± 0.19	0.260	>0.05
CI	1.83 ± 0.10	1.81 ± 0.10	1.77 ± 0.13	0.110	>0.05
ODR	21.32 ± 1.26	21.28 ± 1.44	21.27 ± 0.87	1.177	>0.05
LDR	7.78 ± 0.65	7.82 ± 0.39	7.56 ± 0.58	0.549	>0.05
COX	1.84 ± 0.06	1.84 ± 0.07	1.82 ± 0.06	0.061	>0.05
DFA	0.22 ± 0.02	0.22 ± 0.02	0.23 ± 0.02	0.020	>0.05
OFA	25.11 ± 0.97	25.00 ± 1.00	24.93 ± 0.88	0.926	>0.05

Con—control group; Phy1—control group + phytase 100 g/t; Phy2—control group + phytase 400 g/t; TI—thrombogenicity index; S/P—saturated fat index; OL—oleic acid (C18:1); LA—linoleic acid (C18:2); ALA—alpha linoleic acid (C18:3); CI—consumer index; ODR—desaturation ratio from oleic to linoleic acid; LDR—desaturation ratio from linoleic to linolenic acid; COX—calculated oxidizable value; DFA—index of desirable fatty acids; OFA—sum of hypercholesterolemic fatty acids.

**Table 8 life-13-01275-t008:** Bone mineral analysis.

Variable	Con	Phy1	Phy2	SEM	*p*
Ash (%)	52.47 ± 3.07	53.90 ± 2.60	52.60 ± 2.31	0.447	>0.05
Ca (%)	33.89 ± 1.38	33.27 ± 1.19	33.25 ± 1.46	0.050	>0.05
P (%)	16.71 ^a^ ± 0.24	17.28 ^b^ ± 0.13	17.28 ^b^ ± 0.12	0.224	<0.0001

Con—control group; Phy1—control group + phytase 100 g/t; Phy2—control group + phytase 400 g/t. ^a^,^b^ Means with different superscripts in a row are different (*p* ≤ 0.05).

## Data Availability

Data are available at a reasonable request to the corresponding authors.

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
