# Peer review of "Phytase Supplementation of Growing-Finishing Pig Diets with Extruded Soya Seeds and Rapeseed Meal Improves Bone Mineralization and Carcass and Meat Quality"

_life, 2023, doi:10.3390/life13061275_

Round 1
Reviewer 1 Report
The title of the work focuses on the productive response (performance) and there is only one pen for each treatment, so it should focus on mineralization and carcass and meat and blood measurements. Since 1 vs. 1 vs. 1 is unreliable. But the rest of the data that is evaluated in each pig is useful for the scientific community. The performance only mention it but not as the main objective due to its poor number of replicas. IT DOES NOT MENTION THE EXPERIMENTAL DESIGN USED IN A SPECIFIC WAY, IT MUST BE CLEAR AND PRECISE. Pigs with phytase increased more than 4 kg of carcass, which is a 4.42% increase in carcass yield and was not detected by the test statistic due to low repeatability. IT IS STRONGLY RECOMMENDED TO CARRY OUT A PROFITABILITY ANALYSIS TO MINIMIZE THE LACK OF AN ADEQUATE PERFORMANCE MEASUREMENT. There are interesting data, so if you assume the corrections it is a work that deserves to be published. HIGHLY RECOMMENDED TO MODIFY THE TITLE TO BE PUBLISHED.Author Response
We would like to thank the Reviewer for the very detailed comments to our manuscript, which greatly helped to improve our review. The resubmitted version of the paper was adjusted according to those suggestions. The manuscript in its revised form has been approved by all authors.
Reviewer 1
- The title of the work focuses on the productive response (performance) and there is only one pen for each treatment, so it should focus on mineralization and carcass and meat and blood measurements. Since 1 vs. 1 vs. 1 is unreliable. But the rest of the data that is evaluated in each pig is useful for the scientific community. The performance only mention it but not as the main objective due to its poor number of replicas. It does not mention the experimental design used in a specific way, it must be clear and precise.
Thank you for your comment. We understand your remark but one replication is for FCR, because body weight gain was controlled individually. In the text is information: Each pig was marked with an earring with a number, which allowed individual measurements. As recommended for reviewer, we changed title, abstract and focused on mineralization and carcass, and meat measurements.
- Pigs with phytase increased more than 4 kg of carcass, which is a 4.42% increase in carcass yield and was not detected by the test statistic due to low repeatability.
The research was carried out in farm conditions where, despite equal starting weights, differences in fattening and post-slaughter effects between individual animals usually occur. For example, similar observations were noted in the other studies:
Sońta, M., Rekiel, A., Więcek, J., & Batorska, M. (2020). Economic efficiency of the production of fatteners fed blue lupine as a replacement for GM soybean meal. DOI: 10.5604/01.3001.0014.2017. Authors used blue lupine as a replacement for GM soybean meal (table 4)
Kasprowicz-Potocka, M., Zaworska, A., Kaczmarek, S. A., & Rutkowski, A. (2016). The nutritional value of narrow-leafed lupine (Lupinus angustifolius) for fattening pigs. Archives of Animal Nutrition, 70(3), 209-223.The nutritional value of narrow-leafed lupine (Lupinus angustifolius) for fattening pigs DOI:j10.1080/1745039X.2016.1150238 (table 8).
- It is strongly recommended to carry out a profitability analysis to minimize the lack of an adequate performance measurement.
Simplified calculation of efficiency of production of fatteners is not possible, because we have not noted the costs incurred for production. Additionally, soybeans and phytase were obtained free of charge, hence the calculation would be underestimated. We added other indicators that we could calculate based on the data we have (table 7).
- There are interesting data, so if you assume the corrections, it is a work that deserves to be published.
Thank you, we change and added some part for reviewer suggestion.
- Highly recommended to modify the title to be published.
Title was changed.
We hope the changes made increased the quality of the manuscript. We would be glad if the manuscript in its revised version could be published in MDPI Animals.
Sincerely,
Anita Zaworska-Zakrzewska
Reviewer 2 Report
Microorganisms 2321772 Peer Review Report: The performance, bone mineralization, and carcass quality of fatteners fed diets with extruded soya seeds and rapeseed meal supplemented with different phytase doses.
Abstract:
The first two sentences are unnecessary at this stage. The abstract should be very attractive and informative to your readers. It should carry a condensed summary of your work…introduction, methodology, results, and conclusion. The abstract should be rewritten and expanded.
Introduction
This is satisfactory.
Line 31. The main factor determining the profitability of pig production is feeding costs; feed costs account for about…% of total variable costs (Reference).
Lines 32-33. A comma (,) is missing after currently. It is equally safer to write that soya is one of the most important sources of protein in animal nutrition worldwide, and about 80% of soybeans grown globally are GMOs [1].
Line 37 Actually and the appear unnecessary in this sentence, consider removing them.
Line 43 The comma after the word according appears to be out of place and own appears redundant, consider removing them.
Line 45 It is better to write the word enzyme in a plural form and add such before as. This will help in making the sentence clearer and more meaningful.
Line 47,48 The word addition appears repeated in this sentence. Consider using one of its synonyms. Suggestion: Complement, expansion.
Lines 52-54. The sentence is not clear. It is necessary to make some adjustments to improve clarity. Suggestion: According to scientific publications, using a high-dose phytase additive in feed mixtures for fatteners with processed plant protein sources has yet to be sufficiently evaluated [15].
Lines 57-59. Appear too long and difficult to read and understand. The sentence could be broken into two. Suggestion: Cauble et al. [16] and Gebert et al. [17] reported that dietary phytase could modulate muscle fatty acid profiles in chickens and pigs. Still, other authors need to prove those results.
Lines 63-65. The sentence is unclear and hard to follow, consider rephrasing it. Suggestion: The study aimed to determine how a standard amount of phytase and a higher dosage added to a diet containing RSM and FFES influenced the performance of fatteners and their slaughter efficiency, as well as bone mineralization and meat quality.
Materials and methods
Line 90 were and the appear unnecessary here, consider removing them.
Line 97 Consider changing was to the plural form (were) to make the sentence clearer.
Line 100 Write source in plural form (sources) to make the sentence clearer.
Results
These were well presented except for some few observations below. Lines 181, 301, 304 et al., may have been used incorrectly. Remove the comma (,).
Line 182 The verb pre-process is incorrectly used, consider changing it. Suggestion: pre-processed.
Line 220-221 The sentence hard to follow, consider recasting it. Suggestion: One-way ANOVA only showed a tendency (p = 220 0.054) of the impact of the phytase on the pork's yellow colour (b*).
Line 231 as compared with makes the sentence hard to follow, consider rephrasing these words with “compared to”.
Line 234-235 Your sentence is hard to follow. Think about rephrasing it. Suggestion: The total MUFA amounted to about 47 %, whereas SFA amounted to about 39% and PUFA 14% (Table 5).
Line 261 It appears that the form of the verb design does not work with is in this sentence. Suggestion: is not designed. The word typical needs a determiner a (a typical).
Line 265 Were seems to be in the wrong tense. Suggestion: are
Line 274 The singular countable noun function follows the quantifier both, which requires a plural noun, functions.
Line 276 p value is missing a hyphen, consider adding it.
Line 278 Cross validation seems to be missing a hyphen, consider adding it.
Discussion
Fantastic results were obtained for meat quality traits especially fatty acid profile and meat pH. These were loosely discussed. To improve the quality of the paper, there is need to consult some recent literature sources to rewrite this section. The following are suggested to accomplish this task:
1. Khadre, A. A. B. A., & Karabacak, A. (2018). Comparison of fattening performance and carcass traits measurements of Akkaraman and Awassi male lambs. Selcuk Journal of Agriculture and Food Sciences, 32(3), 542-548.
2. Vahedi, V., Hedayat-Evrigh, N., Holman, B. W., & Ponnampalam, E. N. (2021). Supplementation of macro algae (Azolla pinnata) in a finishing ration alters feed efficiency, blood parameters, carcass traits and meat sensory properties in lambs. Small Ruminant Research, 203, 106498.
3. Pewan, S. B., Otto, J. R., Kinobe, R. T., Adegboye, O. A., & Malau-Aduli, A. E. O. (2021). Nutritional enhancement of health beneficial omega-3 long-chain polyunsaturated fatty acids in the muscle, liver, kidney, and heart of Tattykeel Australian White MARGRA lambs fed pellets fortified with omega-3 oil in a feedlot system. Biology, 10(9), 912.
4. Van Le, H., Nguyen, D. V., Vu Nguyen, Q., Malau-Aduli, B. S., Nichols, P. D., & Malau-Aduli, A. E. O. (2019). Fatty acid profiles of muscle, liver, heart and kidney of Australian prime lambs fed different polyunsaturated fatty acids enriched pellets in a feedlot system. Scientific Reports, 9(1), 1238.
5. Holman, B. W., Kerr, M. J., Refshauge, G., Diffey, S. M., Hayes, R. C., Newell, M. T., & Hopkins, D. L. (2021). Post-mortem pH decline in lamb semitendinosus muscle and its relationship to the pH decline parameters of the longissimus lumborum muscle: A pilot study. Meat Science, 176, 108473.
6. Pewan, S. B., Otto, J. R., Kinobe, R. T., Adegboye, O. A., & Malau-Aduli, A. E. O. (2020). MARGRA lamb eating quality and human health-promoting omega-3 long-chain polyunsaturated fatty acid profiles of Tattykeel Australian White Sheep: Linebreeding and gender effects. Antioxidants, 9(11), 1118.
7. Chiofalo, V., Liotta, L., Lo Presti, V., Gresta, F., Di Rosa, A. R., & Chiofalo, B. (2020). Effect of dietary olive cake supplementation on performance, carcass characteristics, and meat quality of beef cattle. Animals, 10(7), 1176.
8. Stenberg, E., Karlsson, A., Öghren, C., & Segerkvist, K. A. (2020). Carcass characteristics and meat quality attributes in lambs reared indoors, on cultivated pasture, or on semi-natural pasture. Agricultural and Food Science, 29(5), 432-441.
9. Abhijith, A., Warner, R. D., Ha, M., Dunshea, F. R., Leury, B. J., Zhang, M., ... & Chauhan, S. S. (2021). Effect of slaughter age and post-mortem days on meat quality of longissimus and semimembranosus muscles of Boer goats. Meat Science, 175, 108466.
10. Pewan SB, Otto JR, Kinobe RT, Adegboye OA and Malau-Aduli AEO (2022) Fortification of diets with omega-3 long-chain polyunsaturated fatty acids enhances feedlot performance, intramuscular fat content, fat melting point, and carcass characteristics of Tattykeel Australian White MARGRA lambs. Front. Vet. Sci. 9:933038. doi: 10.3389/fvets.2022.933038.
Line 294 The preposition used here may have made the sentence unclear. Consider using on instead of from.
Line 308 The adverbial also appears to be misplaced here. Suggestion: also depends.
Lines 312-315 The sentence is not clearly stated which makes it hard to follow. Consider rephrasing it and breaking it into 2 sentences. Suggestion: In this study, lower FCR in the whole experiment was probably the result of higher energy availability due to the decomposition of phytate complexes with some nutrients. This was also proved by the significantly higher deposition of phosphorus in the bones of pigs offered enzyme [29].
Line 334 The word elements may be unnecessary, consider removing it.
Line 336 It seems that conjunction use may be incorrect here, replace what’s with what. It also seems that the verb affect does not agree with the subject, consider replacing with affects.
Line 350 The singular verb claims does not.
Line 356 It appears that they become may be unnecessary in this sentence. You may consider removing it.
Conclusion
This requires some amendments and improvement. Nothing is mentioned about fatty acid composition. A sentence or two should be devoted for this.
Line 360 The quantifier use may be incorrect here. Suggestion: Add the word such before as.
Lines 360-362 The sentence is too long and difficult to follow. Breaking it into two sentences could help in conveying the idea through. Suggestion: In summary, adding phytase may benefit the primary dose in diets containing processed protein raw materials such as RSM and full-fat soybean seeds. However, it did not enhance the performance of fattening pigs, nor did it result in higher carcass or meat quality.
Line 365. Similarly, can better be written as similar and farms should be written in plural form (farms).
Line 366. There seems to be a word problem here. Suggestion: use slightly reduce not reduce slightly. Cost, better written or used as costs.

Author Response
We would like to thank the Reviewer for the very detailed comments to our manuscript, which greatly helped to improve our review. The resubmitted version of the paper was adjusted according to those suggestions. The manuscript in its revised form has been approved by all authors.
Microorganisms 2321772 Peer Review Report: The performance, bone mineralization, and carcass quality of fatteners fed diets with extruded soya seeds and rapeseed meal supplemented with different phytase doses.
Abstract:
- The first two sentences are unnecessary at this stage. The abstract should be very attractive and informative to your readers. It should carry a condensed summary of your work…introduction, methodology, results, and conclusion. The abstract should be rewritten and expanded.
The abstract was rewritten and expanded. Lines 17-32
- Line 31. The main factor determining the profitability of pig production is feeding costs; feed costs account for about…% of total variable costs (Reference).
Text and reference added in section. Line 38
- Lines 32-33. A comma (,) is missing after currently. It is equally safer to write that soya is one of the most important sources of protein in animal nutrition worldwide, and about 80% of soybeans grown globally are GMOs [1].
It was improved.
- Line 37 Actually and the appear unnecessary in this sentence, consider removing them. Line 43 The comma after the word according appears to be out of place and own appears redundant, consider removing them.
It was improved.
- Line 45 It is better to write the word enzyme in a plural form and add such before as. This will help in making the sentence clearer and more meaningful.
It was changed.
- Line 47,48 The word addition appears repeated in this sentence. Consider using one of its synonyms. Suggestion: Complement, expansion.
It was changed.
- Lines 52-54. The sentence is not clear. It is necessary to make some adjustments to improve clarity. Suggestion: According to scientific publications, using a high-dose phytase additive in feed mixtures for fatteners with processed plant protein sources has yet to be sufficiently evaluated [15].
It was changed.
- Lines 57-59. Appear too long and difficult to read and understand. The sentence could be broken into two. Suggestion: Cauble et al. [16] and Gebert et al. [17] reported that dietary phytase could modulate muscle fatty acid profiles in chickens and pigs. Still, other authors need to prove those results.
It was changed for reviewer suggestion.
- Lines 63-65. The sentence is unclear and hard to follow, consider rephrasing it. Suggestion: The study aimed to determine how a standard amount of phytase and a higher dosage added to a diet containing RSM and FFES influenced the performance of fatteners and their slaughter efficiency, as well as bone mineralization and meat quality.
It was changed for reviewer suggestion.
- Line 90 were and the appear unnecessary here, consider removing them.
It was changed.
- Line 97 Consider changing was to the plural form (were) to make the sentence clearer.
It was changed.
- Line 100 Write source in plural form (sources) to make the sentence clearer.
It was changed.
- These were well presented except for some few observations below. Lines 181, 301, 304 et al., may have been used incorrectly. Remove the comma (,).
It was changed.
- Line 182 The verb pre-process is incorrectly used, consider changing it. Suggestion: pre-processed.
It was changed.
- Line 220-221 The sentence hard to follow, consider recasting it. Suggestion: One-way ANOVA only showed a tendency (p = 220 0.054) of the impact of the phytase on the pork's yellow colour (b*).
It was changed.
- Line 231 as compared with makes the sentence hard to follow, consider rephrasing these words with “compared to”.
It was changed.
- Line 234-235 Your sentence is hard to follow. Think about rephrasing it. Suggestion: The total MUFA amounted to about 47 %, whereas SFA amounted to about 39% and PUFA 14% (Table 5).
It was changed.
- Line 261 It appears that the form of the verb design does not work with is in this sentence. Suggestion: is not designed. The word typical needs a determiner a (a typical).
It was changed.
- Line 265 Were seems to be in the wrong tense. Suggestion: are
It was changed.
- Line 274 The singular countable noun function follows the quantifier both, which requires a plural noun, functions.
It was changed.
- Line 276 p value is missing a hyphen, consider adding it.
It was added.
- Line 278 Cross validation seems to be missing a hyphen, consider adding it.
It was added.
- Fantastic results were obtained for meat quality traits especially fatty acid profile and meat pH. These were loosely discussed. To improve the quality of the paper, there is need to consult some recent literature sources to rewrite this section. The following are suggested to accomplish this task:
- Khadre, A. A. B. A., & Karabacak, A. (2018). Comparison of fattening performance and carcass traits measurements of Akkaraman and Awassi male lambs. Selcuk Journal of Agriculture and Food Sciences, 32(3), 542-548.
- Vahedi, V., Hedayat-Evrigh, N., Holman, B. W., & Ponnampalam, E. N. (2021). Supplementation of macro algae (Azolla pinnata) in a finishing ration alters feed efficiency, blood parameters, carcass traits and meat sensory properties in lambs. Small Ruminant Research, 203, 106498.
- Pewan, S. B., Otto, J. R., Kinobe, R. T., Adegboye, O. A., & Malau-Aduli, A. E. O. (2021). Nutritional enhancement of health beneficial omega-3 long-chain polyunsaturated fatty acids in the muscle, liver, kidney, and heart of Tattykeel Australian White MARGRA lambs fed pellets fortified with omega-3 oil in a feedlot system. Biology, 10(9), 912.
- Van Le, H., Nguyen, D. V., Vu Nguyen, Q., Malau-Aduli, B. S., Nichols, P. D., & Malau-Aduli, A. E. O. (2019). Fatty acid profiles of muscle, liver, heart and kidney of Australian prime lambs fed different polyunsaturated fatty acids enriched pellets in a feedlot system. Scientific Reports, 9(1), 1238.
- Holman, B. W., Kerr, M. J., Refshauge, G., Diffey, S. M., Hayes, R. C., Newell, M. T., & Hopkins, D. L. (2021). Post-mortem pH decline in lamb semitendinosus muscle and its relationship to the pH decline parameters of the longissimus lumborum muscle: A pilot study. Meat Science, 176, 108473.
- Pewan, S. B., Otto, J. R., Kinobe, R. T., Adegboye, O. A., & Malau-Aduli, A. E. O. (2020). MARGRA lamb eating quality and human health-promoting omega-3 long-chain polyunsaturated fatty acid profiles of Tattykeel Australian White Sheep: Linebreeding and gender effects. Antioxidants, 9(11), 1118.
- Chiofalo, V., Liotta, L., Lo Presti, V., Gresta, F., Di Rosa, A. R., & Chiofalo, B. (2020). Effect of dietary olive cake supplementation on performance, carcass characteristics, and meat quality of beef cattle. Animals, 10(7), 1176.
- Stenberg, E., Karlsson, A., Öghren, C., & Segerkvist, K. A. (2020). Carcass characteristics and meat quality attributes in lambs reared indoors, on cultivated pasture, or on semi-natural pasture. Agricultural and Food Science, 29(5), 432-441.
- Abhijith, A., Warner, R. D., Ha, M., Dunshea, F. R., Leury, B. J., Zhang, M., ... & Chauhan, S. S. (2021). Effect of slaughter age and post-mortem days on meat quality of longissimus and semimembranosus muscles of Boer goats. Meat Science, 175, 108466.
- Pewan SB, Otto JR, Kinobe RT, Adegboye OA and Malau-Aduli AEO (2022) Fortification of diets with omega-3 long-chain polyunsaturated fatty acids enhances feedlot performance, intramuscular fat content, fat melting point, and carcass characteristics of Tattykeel Australian White MARGRA lambs. Vet. Sci. 9:933038. doi: 10.3389/fvets.2022.933038.
This part of discussion was improved, but in the current research no differences were found in fatty acids and pH of meat. Only C22 content was lower in the groups with phytase what was supplemented. Some suggested positions were added in this section. Lines: 406-420
- Line 294 The preposition used here may have made the sentence unclear. Consider using on instead of from. Line 308 The adverbial also appears to be misplaced here. Suggestion: also depends.
It was changed.
- Lines 312-315 The sentence is not clearly stated which makes it hard to follow. Consider rephrasing it and breaking it into 2 sentences. Suggestion: In this study, lower FCR in the whole experiment was probably the result of higher energy availability due to the decomposition of phytate complexes with some nutrients. This was also proved by the significantly higher deposition of phosphorus in the bones of pigs offered enzyme [29].
It was changed.
- Line 334 The word elements may be unnecessary, consider removing it.
Line 336 It seems that conjunction use may be incorrect here, replace what’s with what. It also seems that the verb affect does not agree with the subject, consider replacing with affects. Line 350 The singular verb claims does not. Line 356 It appears that they become may be unnecessary in this sentence. You may consider removing it.
It was changed.
- This requires some amendments and improvement. Nothing is mentioned about fatty acid composition. A sentence or two should be devoted for this.
Section conclusions was supplemented. Lins: 440-450
- Line 360 The quantifier use may be incorrect here. Suggestion: Add the word such before as.
It was added in text.
- Lines 360-362 The sentence is too long and difficult to follow. Breaking it into two sentences could help in conveying the idea through. Suggestion: In summary, adding phytase may benefit the primary dose in diets containing processed protein raw materials such as RSM and full-fat soybean seeds. However, it did not enhance the performance of fattening pigs, nor did it result in higher carcass or meat quality.
It was changed.
- Line 365. Similarly, can better be written as similar and farms should be written in plural form (farms). Line 366. There seems to be a word problem here. Suggestion: use slightly reduce not reduce slightly. Cost, better written or used as costs.
It was changed.
We hope the changes made increased the quality of the manuscript. We would be glad if the manuscript in its revised version could be published in MDPI Animals.
Sincerely,
Anita Zaworska-Zakrzewska
Reviewer 3 Report
The subject of this article deals with an interesting and relevant issue question. Although the problem of dietary phytase in pigs has already been addressed from various aspects, this paper approaches the issue from a little-known point of view of meat quality. Unfortunately, I have serious concerns about the title, the methodological quality of the study, and also about the scientific contribution to the interpretation of the results, especially the connection between the use of phytase and meat quality. Moreover, there are too many minor English errors and I recommend the authors check the manuscript in detail. Some of the main concerns are listed below.
Please explain why the starter diet contains wheat and barley but not in grower and finisher. There is a high variation in CP, fat, Ca, and P contents among Con, Phy1, and Phy2. It might be because of the matrix value of phytase? If yes, please remove it. Please clarify.
Table 1, as there is not much difference between Con, Phy1, and Phy2 unless Quantum Blue 5G and corn, I recommend presenting only one diet for each phase (starter, grower, finisher) and adding additional information in the footnote.
Line 23 Please clarify this sentence “The most of the growth 23 performance and meat quality parameters did not differ significantly”
Abstract
Please remove insignificant results, instead please emphasize on significant results such as bone mineralization etc… please have a better conclusion because Phy1 showed several benefits compared with the Con.
Introduction
The introduction gives insufficient information to the readers. Please provide these information:
1- please add the reason why extruded soya seeds and rapeseed meal were used together
2- In line 43-44 its mentioned the stive role of heat processing on phytate degradation which is in contrast with using dietary enzymes. Please clarify.
3- Please add information how phytase supplementation can affect meat quality and fatty acid contents.
4- Please remove “According to scientific publications, the use of a high dose phytase additive in feed mixtures for fatteners with processed plant protein sources has so far been insufficiently evaluated [15].” Because there are so many publications about the role of phytase in fattening pigs even in high doses. Instead, please add further information about the role and importance of phytase.
Results
Please be clear in explaining the results. Line 187, what are “other symptoms”? please add all the checked factors or symptoms.
Please be consistent in putting p values. I recommend removing all the insignificant p-values (p > 0.05). line 196 p = 0.000033 can be changed to p<0.01. etc
Please check all the sentences grammatically. Line 194 please add “the” before “control” and please remove “one”. The control group was already defined as “Con” in line 99. Then its appropriate to use “Con” instead of “control”
Abbreviations. Please check all. For example, body weight gain was abbreviated in line 103 as “BWG”, however, it has not been used anymore in lines 192, 300 etc. Line 39 the “ANF” has not been used throughout the manuscript. Please check all the abbreviations carefully.
Discussion
The discussion has to be improved to discuss the significant results further.
Author Response
We would like to thank the Reviewer for the very detailed comments to our manuscript, which greatly helped to improve our review. The resubmitted version of the paper was adjusted according to those suggestions. The manuscript in its revised form has been approved by all authors.
The subject of this article deals with an interesting and relevant issue question. Although the problem of dietary phytase in pigs has already been addressed from various aspects, this paper approaches the issue from a little-known point of view of meat quality. Unfortunately, I have serious concerns about the title, the methodological quality of the study, and also about the scientific contribution to the interpretation of the results, especially the connection between the use of phytase and meat quality. Moreover, there are too many minor English errors and I recommend the authors check the manuscript in detail. Some of the main concerns are listed below.
- Please explain why the starter diet contains wheat and barley but not in grower and finisher.
In the starter period, animals require better feeds with higher efficiency, while in fattening, special attention is paid to the cost of the mixture. The work was carried out in commercial conditions.
- There is a high variation in CP, fat, Ca, and P contents among Con, Phy1, and Phy2. It might be because of the matrix value of phytase? If yes, please remove it. Please clarify.
Thank you for your comments and remarked. We checked and seen that, in table 1, there were errors in the values for the total protein level in the Phy 2 group. The correct values they were not very different from the values of the mixtures for the Con group. Following the recommendation, we rebuilt Table 1 and we used calculated nutritional value of feeds.
- Table 1, as there is not much difference between Con, Phy1, and Phy2 unless Quantum Blue 5G and corn, I recommend presenting only one diet for each phase (starter, grower, finisher) and adding additional information in the footnote.
In accordance with the recommendation, one column presenting diet for each of the period, and the other information were added in the footnotes.
- Line 23 Please clarify this sentence “The most of the growth performance and meat quality parameters did not differ significantly”
The sentence from the abstract was removed and explained otherwise. Lines: 27-29.
- Please remove insignificant results, instead please emphasize on significant results such as bone mineralization etc… please have a better conclusion because Phy1 showed several benefits compared with the Con.
The abstract was rewritten and expanded. Lines: 17-32.
- The introduction gives insufficient information to the readers. Please provide this information: please add the reason why extruded soya seeds and rapeseed meal were used together. In line 43-44 it is mentioned the stive role of heat processing on phytate degradation which is in contrast with using dietary enzymes. Please clarify.
It was explained in text. Information about hydrolyse of phytate was added. Lines: 53-55.
- Please add information how phytase supplementation can affect meat quality and fatty acid contents.
It was added in text. Lines: 70-72.
- Please remove “Accordingto scientific publications, the use of a high dose phytase additive in feed mixtures for fatteners with processed plant protein sources has so far been insufficiently evaluated [15].” Because there are so many publications about the role of phytase in fattening pigs even in high doses. Instead, please add further information about the role and importance of phytase.
It was removed and added some information about role phytase. Lines: 70-72.
- Results. Please be clear in explaining the results. Line 187, what are “other symptoms”? please add all the checked factors or symptoms.
It was explained in text. Lines: 229-231
- Please be consistent in putting p values. I recommend removing all the insignificant p-values (p> 05). line 196 p = 0.000033 can be changed to p<0.01. etc
It was changed in all tables and text.
- Please check all the sentences grammatically.
One of the reviewers has suggested that manuscript should undergo extensive English revisions, that’s why we sent document to Language Editing Service. All mistake and were corrected in the text and tables.
- Line 194 please add “the” before “control” and please remove “one”. The control group was already defined as “Con” in line 99. Then its appropriate to use “Con” instead of “control”
It was changed in the text.
- Please check all. For example, body weight gain was abbreviated in line 103 as “BWG”, however, it has not been used anymore in lines 192, 300 etc. Line 39 the “ANF” has not been used throughout the manuscript. Please check all the abbreviations carefully.
Checked and corrected all the abbreviations in text.
- The discussion has to be improved to discuss the significant results further.
Discussion was supplemented.
We hope the changes made increased the quality of the manuscript. We would be glad if the manuscript in its revised version could be published in MDPI Animals.
Sincerely,
Anita Zaworska-Zakrzewska
Reviewer 4 Report
Notes to the manuscript:
1. 1. The research methodology does not specify: Ca: P ratio in feed?
2. 2. What was the sex of fattening pigs: females and males or barrows?
3. 3. It is difficult to call young pigs weighing >30 kg “piglets”, these are “weaners”.
4. 4. In the Starter and Grower mixtures for pigs of the Phy2 group, the crude protein level was lowest relative to the crude protein level in the feed for pigs in the other groups, which may have influenced the fattening results.
5. 5. In the section "Carcasses and meat analyses" "lard" is given instead of "backfat thickness".
6. 6. Reference to the publication of Lisiak et al. (2014) regarding the methodology of qualitative meat testing makes it difficult to learn the detailed methodology of the research, although this form is generally accepted.
7. 7. Table 3, however, does not explain what the measurements of backfat thickness at points KI, KII, KIII mean.
8. 8. In Table 2, the results of fattening efficiency are not very transparent. It would be better to give daily weight gains. The obtained total results in FCR of 3.25 kg/kg for group K, respectively 2.96 and 2.98 for groups Phy1 and Phy 2 are too high in relation to the level of FCR in pigs with high potential and should be 2.3-2.5 kg / kg. How did the authors obtain such results?
9. 9. In Table 4 there should be "water holding capacity".
10. 10. Table 6 – how do the Authors explain that the addition of phytase enzymes did not increase the absorption and absorption of Ca?
11. 11. In Table 5, the authors showed that they obtained a normal ratio of PUFA n-6 to PUFA n-3 in meat. It is a pity that they did not calculate the indices: AI and TI and S/P.
Author Response
We would like to thank the Reviewer for the very detailed comments to our manuscript, which greatly helped to improve our review. The resubmitted version of the paper was adjusted according to those suggestions. The manuscript in its revised form has been approved by all authors.
Comments and Suggestions for Authors
Notes to the manuscript:
- The research methodology does not specify: Ca: P ratio in feed?
We added Ca/P in table 1- last row in the table
- What was the sex of fattening pigs: females and males or barrows?
It was clarified in the text: 60 castrated weaners (30 ♀ and 30 ♂) Line: 107
- It is difficult to call young pigs weighing >30 kg “piglets”, these are “weaners”.
Yes, of course, we clarified nomenclature in the text Line:122
- In the Starter and Grower mixtures for pigs of the Phy2 group, the crude protein level was lowest relative to the crude protein level in the feed for pigs in the other groups, which may have influenced the fattening results.
Thank you for your comments and remarked.
We checked and seen that, in table 1, there were errors in the values for the total protein level in the Phy 2 group. The correct values they were not very different from the values of the mixtures for the Con group. Following the recommendation of one of the reviewers, we rebuilt Table 1.
- In the section "Carcasses and meat analyses" "lard" is given instead of "backfat thickness".
It was changed. Line: 152
- Reference to the publication of Lisiak et al. (2014) regarding the methodology of qualitative meat testing makes it difficult to learn the detailed methodology of the research, although this form is generally accepted.
According to the reviewer's suggestion, the methodology for meat quality assessment was precisely described. Lines: 149-165
- Table 3, however, does not explain what the measurements of backfat thickness at points KI, KII, KIII mean.
It was added under table 3 - Backfat thickness were measured 3 points KI-KIII (KI- less than 22 mm, KII- 22-26 mm and over 26mm- KIII). Lines: 152-153
- In Table 2, the results of fattening efficiency are not very transparent. It would be better to give daily weight gains. The obtained total results in FCR of 3.25 kg/kg for group K, respectively 2.96 and 2.98 for groups Phy1 and Phy 2 are too high in relation to the level of FCR in pigs with high potential and should be 2.3-2.5 kg / kg. How did the authors obtain such results?
Yes, daily body weight gain is better and practically than BWG/ transparent. We were changed BWG to DWBG in table and text.
In our results we observed high FCR in the Finisher period.
Probably the obtained result was due to 2 factors: We ended the experiment when more than half of the animals reached the slaughter weight of pigs - approx. 125 kg. It is known that with a body weight between 90 and 125 kg, intensive fat synthesis and a slower growth rate is found, but a high feed intake.
In addition, as stated in the M&M section, the studies were conducted under production standard (individual farm) and the animals were kept in group pens, that’s why there may were greater feed losses when spilling out of the feeder.
Our other studies, as well as those of other teams, show similar results that we obtained in this experiment (list below).
We know that, the reduction of FCR makes financial sense and supports sustainable production. As consumers and policy continue to highlight the need for sustainable production, the use of FCR as a proxy for environmental sustainability will become increasingly useful, that's why we want to stay this results in the manuscript.
Ding, X., Li, H., Wen, Z., Hou, Y., Wang, G., Fan, J., & Qian, L. (2020). Effects of fermented tea residue on fattening performance, meat quality, digestive performance, serum antioxidant capacity, and intestinal morphology in fatteners. Animals, 10(2), 185.
Kasprowicz-Potocka, M., Zaworska-Zakrzewska, A., & Rutkowski, A. (2020). Effect of Phytase on Digestibility and Performance of Growing and Finishing Pigs Fed Diets with Lupins and Rapeseed Meal. J. Agric. Sci. Technol. A, 10, 216-227.
Lagos, L. V., Walk, C. L., Murphy, M. R., & Stein, H. H. (2019). Effects of dietary digestible calcium on growth performance and bone ash concentration in 50-to 85-kg growing pigs fed diets with different concentrations of digestible phosphorus. Animal Feed Science and Technology, 247, 262-272.
- In Table 4 there should be "water holding capacity".
Yes, thank you for your comments. We added in the table 4.
- Table 6 – how do the Authors explain that the addition of phytase enzymes did not increase the absorption and absorption of Ca?
The concept of bioavailability has included not only the elements absorption but also its final effect on organism growth. The bioavailability of minerals was not measured here. Marked calcium and phosphorus were deposited in the bones, which may indirectly indicate an absorption of these minerals and improved P bioavailability. The calcium level was probably eave in the diet for normal minerals deposition, especially as the diet was supplemented with limestone and calcium-phosphate. The ratio of Ca:P is 1,2-1,4:1.
–
It was added in conclusion. Lines: 448-449.
- In Table 5, the authors showed that they obtained a normal ratio of PUFA n-6 to PUFA n-3 in meat. It is a pity that they did not calculate the indices: AI and TI and S/P.
We added suggested indices.
M&M lines: 195-206, Results: 293-298
We didn't calculate the index because we don't have information about 12:0 acid (formula: (C12:0+4 × C14:0 + C16:0)/(MUFA + PUFA)
We hope the changes made increased the quality of the manuscript. We would be glad if the manuscript in its revised version could be published in MDPI Animals.
Sincerely,
Anita Zaworska-Zakrzewska
Round 2
Reviewer 2 Report
The title of this article should be looked at, and a few adjustments are needed. Suggestion: Phytase supplementation of growing-finishing pig diets with extruded soya seeds and rapeseed meal improves bone mineralization, carcass and meat quality.
A substantial level of implementation of suggestions in the first draft was carried out.
Author Response
Dear Reviewer
Thank you for your suggestion for the title of the article. We were change. In addition, we analyzed the text step by step and made improvements. All correct using the function “Track Changes”. We hope the changes made increased the quality of the manuscript. We would be glad if the manuscript in this version could be published in MDPI Microorganism.
Sincerely,
Anita Zaworska-Zakrzewska
